# Prevalence and Antibiotic Resistance in *Campylobacter* spp. Isolated from Humans and Food-Producing Animals in West Africa: A Systematic Review and Meta-Analysis

**DOI:** 10.3390/pathogens11020140

**Published:** 2022-01-24

**Authors:** Ellis Kobina Paintsil, Linda Aurelia Ofori, Sarah Adobea, Charity Wiafe Akenten, Richard Odame Phillips, Oumou Maiga-Ascofare, Maike Lamshöft, Jürgen May, Kwasi Obiri Danso, Ralf Krumkamp, Denise Dekker

**Affiliations:** 1Kumasi Centre for Collaborative Research in Tropical Medicine (KCCR), South-End, Asuogya Road, Kumasi 039-5028, Ghana; danquah@kccr.de (C.W.A.); phillips@kccr.de (R.O.P.); maiga@bnitm.de (O.M.-A.); 2Department of Theoretical and Applied Biology, Kwame Nkrumah University of Science and Technology, Kumasi 039-5028, Ghana; laandoh.cos@knust.edu.gh (L.A.O.); obirid@knust.edu.gh (K.O.D.); 3Department of Emergency Medicine, Komfo Anokye Teaching Hospital, Okomfo Anokye Road, Kumasi 034-9094, Ghana; SAdobea@kathhsp.org; 4Department of Infectious Disease Epidemiology, Bernhard Nocht Institute for Tropical Medicine (BNITM), Bernhard-Nocht-Str. 74, 20359 Hamburg, Germany; lamshoeft@bnitm.de (M.L.); may@bni-hamburg.de (J.M.); dekker@bnitm.de (D.D.); 5German Centre for Infection Research (DZIF), Partner Site Hamburg-Lübeck-Borstel-Riems, 20359 Hamburg, Germany; 6Tropical Medicine II, University Medical Center Hamburg-Eppendorf (UKE), 20251 Hamburg, Germany

**Keywords:** campylobacteriosis, *Campylobacter*, pooled prevalence, food-producing animals, antibiotic resistance, West Africa

## Abstract

*Campylobacter* species are one of the leading causes of gastroenteritis in humans. This review reports on the prevalence and antibiotic resistance data of *Campylobacter* spp. isolated from humans and food-producing animals in West Africa. A systematic search was carried out in five databases for original articles published between January 2000 and July 2021. Among 791 studies found, 38 original articles from seven (41%) out of the 17 countries in West Africa met the inclusion criteria. For studies conducted in food-producing animals, the overall pooled prevalence of *Campylobacter* spp. was 34% (95% CI: 25–45). The MDR prevalence was 59% (95% CI: 29–84) and half (50%, 13/26) of the animal studies had samples collected from the market. The human studies recorded a lower pooled prevalence of *Campylobacter* spp. (10%, 95% CI: 6–17), but a considerably higher rate of MDR prevalence (91%; 95% CI: 67–98). The majority (85%, 11/13) of the human studies took place in a hospital. *Campylobacter jejuni* and *Campylobacter coli* were the most common species isolated from both animals and humans. Our findings suggest that *Campylobacter* spp. is highly prevalent in West Africa. Therefore, improved farm hygiene and ‘One Health’ surveillance systems are needed to reduce transmission.

## 1. Introduction

Animals are natural reservoirs for *Campylobacter* spp. [1], which are among the leading causes of bacterial gastroenteritis in humans, worldwide [2]. Human *Campylobacter* infection is mainly acquired by the consumption of undercooked poultry, livestock, or by direct contact with animals [1]. A significant proportion of the population in Africa keeps livestock and/or poultry [3]. However, these animals are often reared and slaughtered under poor hygienic and sanitary conditions [4] and high frequencies of *Campylobacter* have been reported in animal husbandry. For example, in Nigeria and Côte d’Ivoire, *Campylobacter* were isolated from 82% [5] and 81% [6] of fecal samples from poultry, respectively. There is sufficient evidence that *Campylobacter* found in retail poultry eventually lead to infections in humans [7].

In humans, the species *Campylobacter jejuni* and *coli* are mainly associated with campylobacteriosis followed by *Campylobacter lari* [8]. Dogs and cats are also known to harbor pathogenic *Campylobacter* species which cause infections in humans [9]. Although *Campylobacter* infections are typically self-limiting, in immunocompromised individuals post-infection complications such as reactive arthritis (painful inflammation of the joints) and Guillain-Barré syndrome (neurological disorders) might occur [10]. Additionally, *Campylobacter* species resistant to commonly used antibiotics are on the increase in Sub-Saharan Africa [11] and worldwide [12]. While information on *Campylobacter* spp. from industrialized countries is broadly available [1], only few meta-analyses have been performed on *Campylobacter* prevalence studies from Africa [8,13,14].

This systematic review and meta-analysis reports on the prevalence and antibiotic resistance data of *Campylobacter* spp. in humans and food-producing animals in West Africa from 1 January 2000 to 31 July 2021. To the best of our knowledge, this is the first meta-analysis from West Africa conducted on *Campylobacter* in both animals and humans in the last 21 years.

## 2. Results

### 2.1. Literature Search

A total of 791 studies were initially identified across Medline (via PubMed), Directory of Open Access Journals (DOAJ), Google Scholar, African Index Medicus and the African Journal Online (AJOL) database. After the removal of duplicate articles, 632 unique articles remained, out of which, 66 articles fulfilled the eligibility criteria for full-text review. Twenty-eight of the full-text articles were excluded due to the following reasons: 16 were not within the scope of this review, five studies did not provide *Campylobacter* prevalence, another five had unclear data, and two were unavailable. Finally, 38 original research articles describing *Campylobacter* prevalence in West African countries were found to be eligible for further analysis. Figure 1 shows a flowchart of the article selection process. Out of the 38 eligible studies, 26 were conducted in food-producing animals [5,6,15,16,17,18,19,20,21,22,23,24,25,26,27,28,29,30,31,32,33,34,35,36,37,38]. One out of the 26 animal studies also had prevalence data on humans [38], thus 13 human studies were included [38,39,40,41,42,43,44,45,46,47,48,49,50]. Four studies conducted in humans were case-control studies and reported *Campylobacter* frequencies in both diarrhea and non-diarrhea patients [39,41,43,48]. One study each was conducted among pig farmers [38] and patients with urinary tract infections (UTIs) [46].

### 2.2. Number of Campylobacter Prevalence Studies Conducted 

*Campylobacter* spp. prevalence data were available from seven (Nigeria, Ghana, Burkina Faso, Côte d’Ivoire, Benin, Niger and The Gambia) out of the 17 countries in West Africa (Figure 2). Approximately 39% (*n*/N = 15/38) of the 38 studies identified in this review were conducted in Nigeria followed by studies conducted in Ghana (24%, *n*/N = 9/38). The majority (87%, *n*/N = 33/38) of the studies were conducted between 2011 and 2021.

### 2.3. Subgroup Analysis of Campylobacter Studies in Animals

Figure 3 shows a forest plot with individual, subgroup and overall pooled prevalence estimates of *Campylobacter* spp. from the 26 animal studies with a total sample size of 9021. The individual prevalence estimates ranged widely, from 4% to 88% in poultry and 11% to 93% in livestock. For the subgroup analysis, poultry recorded a higher pooled prevalence (39%, 95% CI: 27–52) than livestock (26%, 95% CI: 17–38). The overall random-effects pooled prevalence of *Campylobacter* spp. isolated from poultry and livestock samples from West Africa was 34% (95% CI: 25–45) with a very high level of heterogeneity (*I*^2^ = 99%). Figure 4 shows a funnel plot with asymmetric distribution of *Campylobacter* studies conducted in food-producing animals in West Africa. Only three [22,23,37] out of the 26 food-producing animal studies lie within the triangular region, where 95% of the studies are expected to lie. The poultry studies show high variability in the prevalence rates, irrespective of the study sample size. A similar result is observed in the livestock studies; apart from one livestock study [38] that recorded the highest prevalence (93%), all the other seven studies had a prevalence of ≤31% and standard errors of ≤0.03.

Table 1 shows subgroup analysis of *Campylobacter* studies conducted in food-producing animals in West Africa. Three studies from Côte d’Ivoire [6,26,28] recorded the highest country-level pooled prevalence of 74% (95% CI: 52–88) but their combined sample size of 791 (weight = 11.6%) was the third largest. Nigeria, on the other hand, recorded the highest number of studies included [5,19,21,25,27,30,31,32,33,34,36,37,38] with a total sample size of 5702 (weight = 50.5%) and a pooled *Campylobacter* prevalence of 37% (95% CI: 25–51). There were six studies from Ghana [16,17,18,20,24,35] with a sample size of 1917 (weight = 22.8%) and pooled prevalence of 21% (95% CI: 14–30). About 43% (*n*/N = 13/30) of the animal studies collected their samples from markets [6,16,17,18,20,21,22,23,26,28,29] and in these studies, the highest pooled *Campylobacter* prevalence was observed (37%, 95% CI: 23–52). Most of the samples used were carcasses (43%, *n*/N = 13/30), followed by rectal swabs (23%, *n*/N = 7/30) and feces (23%, *n*/N = 7/30). Five studies conducted in animals combined culture and PCR diagnostic methods for the detection of *Campylobacter* [6,22,26,28,34] and this diagnostic method recorded the highest pooled isolation rate of 54% (95% CI: 28–78). The majority (62%, *n*/N = 16/26) of the studies used both culture and biochemical methods for strain identification, which recorded a pooled prevalence of 32% (95% CI: 21–47). Out of the 26 *Campylobacter* studies conducted in animals, 25 (96%) reported data on the various *Campylobacter* species isolated. *C. jejuni* (88%, *n*/N = 22/25) and *C. coli* (68%, *n*/N = 17/25) were the most reported *Campylobacter* spp. with a pooled prevalence of 52% (95% CI: 42–63) and 30% (95% CI: 22–40), respectively.

### 2.4. Subgroup Analysis of Campylobacter Studies in Humans

Thirteen articles on *Campylobacter* prevalence in humans were included, with a total sample size of 6840. The second highest prevalence was observed in a single study [38] conducted among pig farmers and their household members (63%, 95% CI: 54–70). One study [46] conducted among patients with UTI reported a prevalence of 12% (95% CI: 6–22). Figure 5 shows the forest plot of the remaining *Campylobacter* prevalence studies conducted in HIV, diarrhea and non-diarrhea patients. A high pooled prevalence of 45% (95% CI: 19–74) was estimated for HIV patients. The pooled prevalence for diarrhea patients (5%, 95% CI: 3–10) was higher than in non-diarrhea patients (2%, 95% CI: 0–19). The overall pooled estimate in humans was 10% (95% CI: 6–17) with a considerably high level of heterogeneity (*I*^2^ = 98%). Figure 6 shows a funnel plot with asymmetric distribution of subgroups of *Campylobacter* prevalence studies conducted in humans. The majority of studies (61%, *n*/N = 11/18) had low standard errors (≤0.02) and 83% (*n*/N = 15/18) recorded a prevalence of ≤20%. A study [50] conducted among HIV patients had the highest prevalence (68%) as well as the highest standard error (>0.04).

Table 2 shows subgroup analysis of *Campylobacter* studies conducted in humans in West Africa. Most (85%, *n*/N = 11/13) of the studies were conducted in a hospital setting [39,41,42,43,44,45,46,47,48,49,50] and four (29%, *n*/N = 4/14) were diarrhea and non-diarrhea case-control studies [39,41,43,47]. The highest pooled *Campylobacter* prevalence (33%, 95% CI: 13–62) was estimated in adults (>15 years), while children less than five years had the lowest (4%, 95% CI: 2–8). A combination of culture and biochemistry diagnostic methods was used by 46% (*n*/N = 6/13) of the studies to isolate and identify the bacteria [38,42,45,46,48,50]. This method recorded the highest pooled prevalence estimate of 22% (95% CI: 7–51). Eight studies reported data on *Campylobacter* species [38,39,46,47,48,49,50], of which *C. coli* and *C. jejuni* were most common with a pooled prevalence of 47% (95% CI: 25–69) and 42% (95% CI 26–59), respectively.

### 2.5. Antimicrobial Resistance Profile of Campylobacter Species

Apart from ciprofloxacin and nalidixic acid, a higher proportion of antibiotic resistance was observed in *Campylobacter* isolates from humans than from animals (Figure 7). The majority of *Campylobacter* isolates recovered from animals were susceptible to gentamicin (94%, *n*/N = 1183/1265) and those from humans to ciprofloxacin (69%, *n*/N = 175/255).

Resistance to three or more antibiotics, multi-drug resistance (MDR), was reported in 10 studies, two studies in humans [46,50] and eight in animals [6,20,22,24,25,26,35,36]. The overall pooled prevalence estimate for AMR was 69% (95% CI: 40–88, *I*^2^ = 98%), 91% (95% CI: 67–98, *I*^2^ = 65%) in the human studies and 59% (95% CI: 29–84, *I*^2^ = 98%) in food-producing animals. Two studies tested 306 *Campylobacter* isolates against imipenem and observed 0% resistance [20,35]. The following virulence makers: *cdtA, cdtB, cdtC, cadF* [6,26]; antibiotic resistance genes: *tet (O)*, *blaOXA-61*, *aadE*, and *cmeB* [49] and change in amino acid sequence of the *gyrA* gene [16] have been reported by some studies.

## 3. Discussion

This review shows that the majority (58.8%) of countries in West Africa has no published studies on *Campylobacter* prevalence that met our inclusion criteria. This finding is in agreement with other systematic reviews that also report low numbers of *Campylobacter* research in Africa [8,14]. The cumbersome procedures involved in isolating *Campylobacter* spp. makes it difficult for most low-income countries to conduct such studies. Nigeria (50%) and Ghana (23%) had the highest number of studies, probably because these countries have a higher socio-economic status in the region, hence they can afford well-equipped health and research facilities needed to conduct such research [51]. The number of studies published within 2011–2021 was far more than during the preceding decade. This could be because knowledge of new and advanced methods of detection (such as PCR) became available in recent years. Additionally, researchers are now becoming more aware of the burden of *Campylobacter* infections in humans and animals.

Poultry and livestock samples recorded the highest pooled prevalence of *Campylobacter* spp. The intestinal tract of poultry and livestock are frequently colonized in high numbers by *Campylobacter* spp., hence constituting a natural reservoir and an important source of transmission [1,52]. Studies conducted on *Campylobacter* spp. colonization in poultry and livestock are in agreement with the current findings [1,53]. This suggests that poultry and other animals are primary reservoirs responsible for *Campylobacter* infections in humans [54,55].

Our review shows that the pooled prevalence of *Campylobacter* infections in humans was 10% (95% CI: 6–17). Summarized findings from Ethiopia [56] and Sub-Saharan African countries [8] have reported a similar pooled prevalence in humans (9–10%). West Africa has just four published articles on *Campylobacter* infection in children under 5 years of age. Surprisingly, these studies recorded a pooled prevalence of 4%, lower than the 10% reported in a summary of results from Ethiopia [57]. The lower *Campylobacter* prevalence observed in children might be because most of the included studies were conducted in healthy (non-diarrhea) subjects. The low number of studies conducted within this age group shows that some populations in West Africa have not been investigated; hence, there is the need to conduct more studies in these populations. Our review reported high heterogeneity between studies, which could be due to differences in environmental conditions, socio-demographics, sociocultural factors and disease awareness levels. Additionally, the protocols used and the experience level of staff in isolating the bacteria could account for the differences in prevalence observed between studies.

Approximately 56% of the studies used culture and biochemical tests to identify *Campylobacter* spp. Other reviews conducted in Africa have also observed that this method is most common for identifying *Campylobacter* in the region [8,11]. However, the culture method has some limitations; environmental stress during sample transportation and processing can make some *Campylobacter* spp. viable but not culturable on media [58], this could lead to lower sensitivity [59]. In our review, the culture and biochemical method produced a high pooled prevalence of *Campylobacter* species. However, these findings must be interpreted with caution because the culture method has lower specificity compared to PCR-based methods. A lot of researchers in West Africa rely on the laborious and time-consuming culture method because their laboratories are not well equipped to use PCR in the diagnoses of *Campylobacter*. This could be a possible reason why fewer studies have been conducted.

*C. jejuni* and *C. coli* were isolated by 28 and 24 studies, respectively, making them the predominantly isolated *Campylobacter* species from both food-producing animals and humans. Other authors have reported similar observations [8,11]. The high numbers of virulence genes associated with *C. jejuni* and *C. coli* [60] possibly make researchers develop research questions focused on discovering more of these species. Additionally, the high prevalence of *C. jejuni* and *C. coli* observed in this study could be attributed to the culture and biochemical test method used for speciation, which is incapable of detecting the lesser-known *Campylobacter* spp. [61]. *Campylobacter* selective media containing antibiotics and higher incubation temperatures does inhibit the growth of some *Campylobacter* species such as *C. upsaliensis and C. lari* [11]. Nonetheless, it is well known that the two *Campylobacter* species most frequently associated with diarrhea in humans are *C. jejuni* and *C. coli* [62].

The 69% AMR recorded by this review shows that *Campylobacter’s* resistance to commonly used antibiotics is widespread in both humans and animals. Among the most tested antibiotics, *Campylobacter* were found to be highly resistant to tetracycline, nalidixic acid, ciprofloxacin, erythromycin, chloramphenicol, ampicillin and gentamycin. Ciprofloxacin resistance rate in animal isolates was higher than humans, even though it is not approved for use in veterinary medicine. Consistent with our findings, 88% of *E. coli* isolated from poultry farms in Ibadan, Nigeria were resistant to ciprofloxacin [63], suggesting that its use in poultry and livestock farming may be on the increase. We also observed high erythromycin resistance in human isolates compared to animals. The high erythromycin resistance observed in humans could be explained by the overuse of azithromycin due to its *low risk of side effects* [64]. The high antimicrobial-resistant rate observed in our study agrees with findings from similar studies conducted in both low and middle-income countries [65] and high-income countries [66] showing an increasing trend of antibiotic resistance in *Campylobacter* spp. The increasing trend might also be due to the extensive use of antimicrobials in animal farming for growth promotion and prophylaxis [67] and the indiscriminate use in humans [68]. Carbapenem resistance is on the increase in Gram-negative bacteria [69]; however, our review observed that all isolates tested against imipenem were susceptible. The high rate of *Campylobacter* susceptibility observed against imipenem might be attributed to it not being authorized for use in animal husbandry [70].

To lower the high *Campylobacter* prevalence and antimicrobial resistance observed in this review, we recommend the appropriate use of antibiotics in human and veterinary medicine, improved hygiene and sanitation practices and the implementation of biosecurity measures in farms [65]. If possible, antimicrobial susceptibility testing should be performed before the administration of antibiotics to humans. Since the virulence and pathogenicity of *Campylobacter* is affected by the genetic variants, we recommend the use of molecular diagnostic methods in addition or as a replacement to the widely used culture method, in order to accurately diagnose infections and to determine the real *Campylobacter* burden [60]. Furthermore, strong commitment from policymakers is needed to implement ‘One Health’ surveillance systems.

Our systematic review and meta-analysis have few potential limitations. The search strategy was limited to only articles published in English, there might be articles published in other languages that were not considered. The analyses were not uniformly spread since data was absent from majority of the countries and most of the studies were conducted between 2011 and 2021. Another limitation of our review is that majority of the studies used culture methods, which is not the preferred method for reporting *Campylobacter* prevalence. We recorded high heterogeneity because studies conducted in different countries and under different conditions were pooled together. Since we only found data from less than half of the countries in West Africa, our findings may not be generalizable to the entire region.

## 4. Materials and Methods

### 4.1. Study Design and Systematic Review Protocol

The protocol of this review is registered at PROSPERO with registration number: CRD 42021260515. This study was conducted according to the Preferred Reporting Items for Systematic Reviews and Meta-Analyses (PRISMA) guidelines [71]. The UN macro-geographical definition of West Africa (https://unstats.un.org/unsd/methodology/m49/ (accessed on 29 September 2021) was used to define West African countries included in this review, namely: Benin, Burkina Faso, Cape Verde, Côte d’Ivoire, Gambia, Ghana, Guinea, Guinea-Bissau, Liberia, Mali, Mauritania, Niger, Nigeria, Saint Helena, Senegal, Sierra Leone and Togo.

### 4.2. Selection Criteria and Literature Search Strategies

A systematic search for original articles covering West African countries and published between January 2000 and July 2021 was conducted using the following databases: Medline (via PubMed), Directory of Open Access Journals (DOAJ), Google Scholar, African Index Medicus and the African Journal Online (AJOL) database. The systematic search of these databases was performed using the search terms listed in the Appendix A. EKP screened the titles and abstracts of all recovered articles. Articles were eligible for full-text review when: (i) they contained data from a West African country, (ii) they were published between January 2000 to July 2021, and (iii) they were written in English. During the full-text review, two authors (EKP, SA) independently assessed the articles to determine if each one met the inclusion criteria. An article was included if it contained primary data, was conducted in food-producing animals and/or humans, and *Campylobacter* prevalence was reported or can be calculated from available information. Articles whose full texts could not be accessed and those with inconsistent results, overlapping or duplicate data were excluded. Additionally, articles that did not report on the age of study participants, type of samples collected and laboratory diagnostic method used were excluded. In case of any disagreement in the review process, a third reviewer (LAO) was available to give a decisive opinion on any unresolved issues.

### 4.3. Data Extraction

For each included original full-length study article, we extracted data on the first author, year of publication, name of the country where the study was conducted, type of food-producing animals sampled, age of human participants, type of samples collected, sample size, study design and study setting. We also collected data on the laboratory diagnostic methods used, *Campylobacter* prevalence, *Campylobacter* spp. isolated, antimicrobials tested and antibiotic resistance.

### 4.4. Risk of Bias Assessment

Conventional funnel plots show inaccurate results when assessing publication bias in systematic reviews on prevalence studies [72]. This is because of the unequal and small sample sizes, high prevalence diversity due to study design differences and zero prevalence which may be recorded in studies. Therefore, we decided to include all studies that met the final inclusion criteria without assessing the risk of publication bias. Nonetheless, funnel plots were plotted to indicate the across-study biases and between-study heterogeneity.

### 4.5. Data Analysis

In studies that did not explicitly report *Campylobacter* prevalence, but reported the number of positives and the total number of samples collected, the prevalence was calculated as the fraction of both terms. The Meta (version 5.0-1, R Core Team, Vienna, Austria) package, in R software (version 4.1.1, R Core Team, Vienna, Austria) [73], was used to calculate pooled prevalence estimates using a random-effects model [74]. The pooled prevalence with a 95% confidence interval (CI) was presented using forest plots and tables. The heterogeneity of study prevalence estimates was evaluated by computing the inverse variance index (*I*^2^) statistic. Heterogeneity was considered to be high when *I*^2^, which describes the percentage of total variation between studies that is attributable to prevalence differences rather than chance, was above 75%.

Subgroup analyses were used to investigate potential associations with the prevalence estimates. The potential sources of heterogeneity were investigated considering the year of publication, the country where sampling occurred, study setting, sample type, age of human participants, laboratory diagnostic method used and types of *Campylobacter* species isolated. For human studies, the subgroup analysis also included patients with and without diarrhea, HIV and urinary tract infections (UTI). The proportion of *Campylobacter* spp. that were resistant to commonly tested antibiotics was calculated for both food-producing animal and human studies. The ggplot2 package, in the R (version 4.1.1) statistical environment, was used to plot a bar chart to illustrate the proportion of resistant *Campylobacter* in humans and animals. QGIS software (version 3.18.3, QGIS Development Team, Zurich, Switzerland) [75] was used to draw a map to show the number of *Campylobacter* prevalence studies across West Africa.

## 5. Conclusions

Research articles on *Campylobacter* prevalence were not available from 59% of countries in West Africa. Countries in West Africa should be supported to have well-equipped laboratories for *Campylobacter* research. To curb the high *Campylobacter* prevalence and resistance observed in this review, routine diagnosis, appropriate use of antibiotics, improved hygienic practices and ‘One Health’ surveillance systems should be implemented. Furthermore, strong commitment from policymakers and societal actions are needed to improve farm hygiene and antimicrobial usage in food-producing animals and humans.

## Figures and Tables

**Figure 1 pathogens-11-00140-f001:**
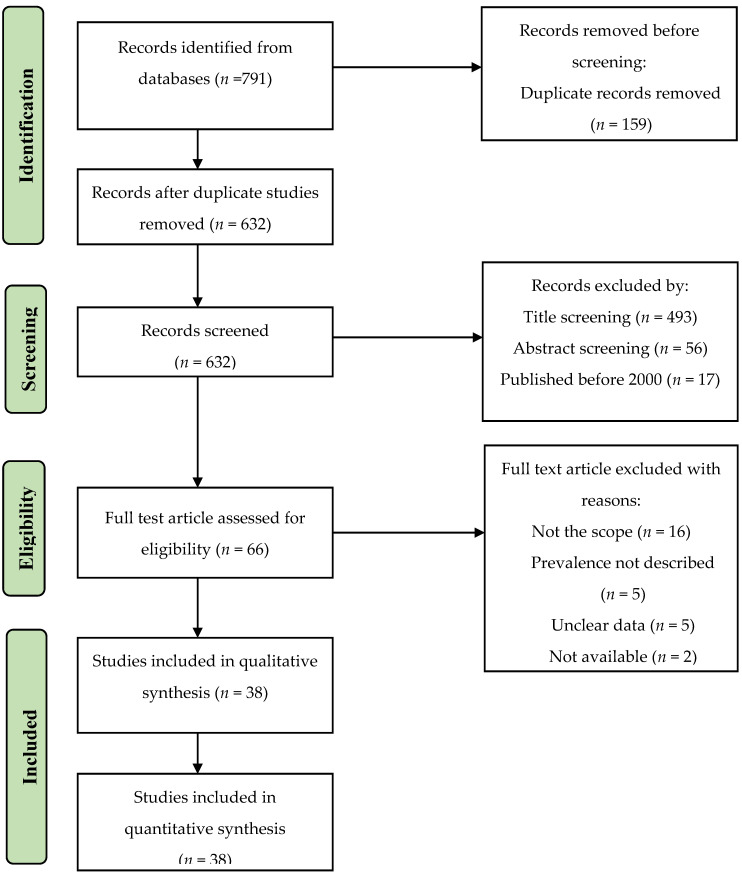
Flow diagram of the article selection process.

**Figure 2 pathogens-11-00140-f002:**
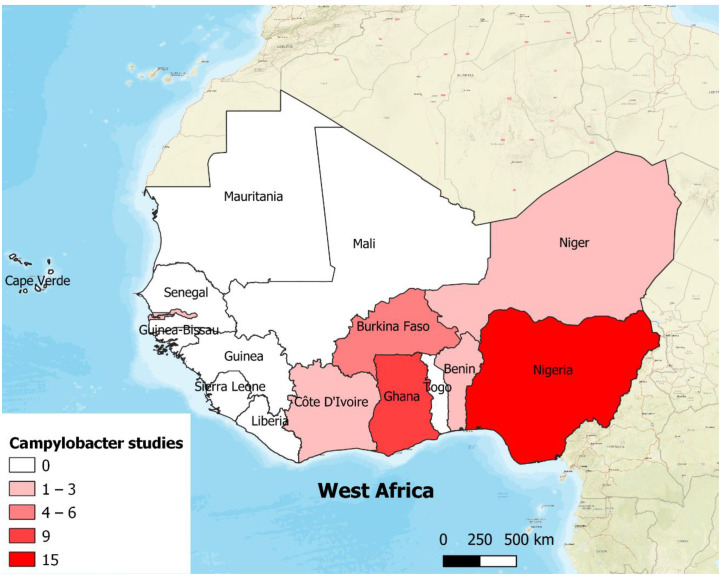
Number of included *Campylobacter* prevalence studies conducted by countries in West Africa between 2000 and 2021.

**Figure 3 pathogens-11-00140-f003:**
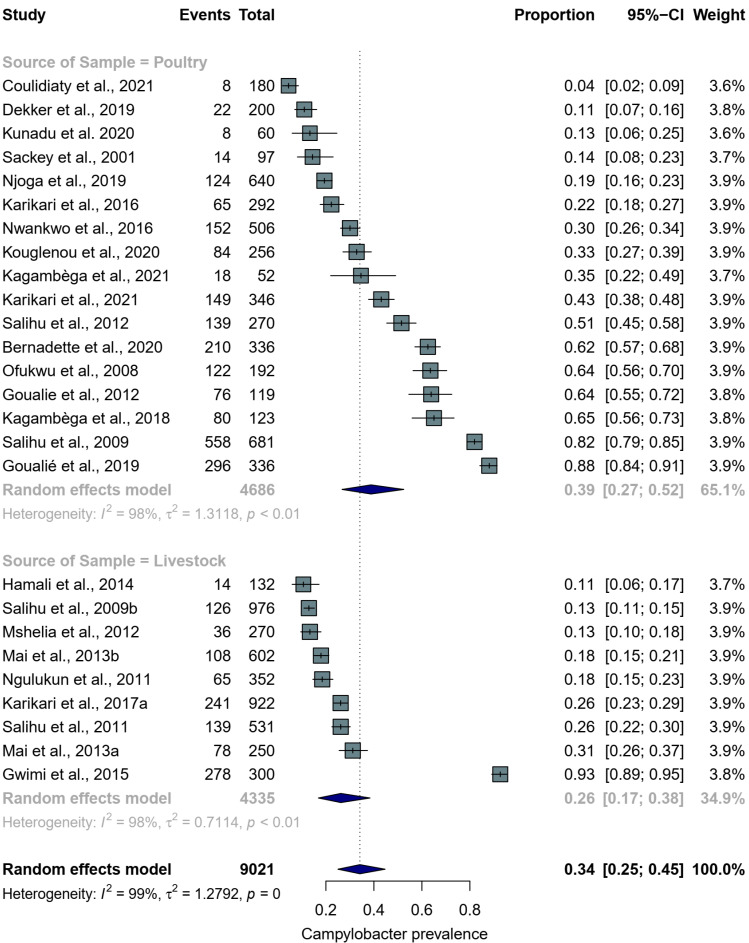
Forest plot showing *Campylobacter* prevalence from poultry and livestock studies from West Africa between 2000 and 2021. The light blue squares represent individual study weight in the meta-analysis and the black lines within the square reflect the 95% CI. The navy blue diamonds represent the results for random effects models, the left and right endpoints of which are the lower and upper bounds of the 95% CI, respectively.

**Figure 4 pathogens-11-00140-f004:**
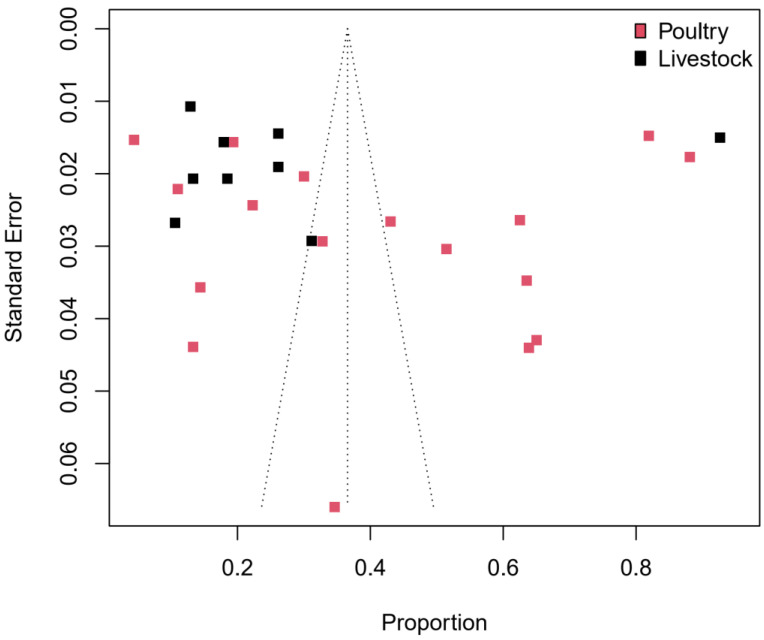
Funnel plot with 95% confidence limits showing the prevalence of *Campylobacter* species in poultry and livestock in West Africa.

**Figure 5 pathogens-11-00140-f005:**
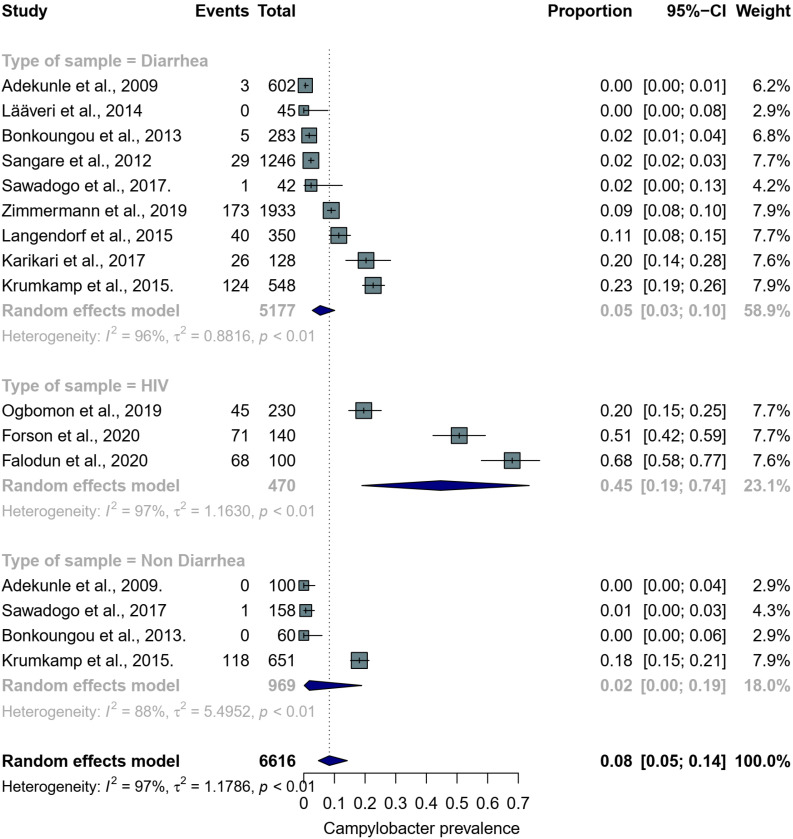
Forest plot showing *Campylobacter* prevalence in HIV, diarrhea and non-diarrhea patients. The light blue squares represent individual study weight in the meta-analysis and the black lines within the square reflect the 95% CI. The navy blue diamonds represent the results for random effects models, the left and right endpoints of which are the lower and upper bounds of the 95% CI, respectively.

**Figure 6 pathogens-11-00140-f006:**
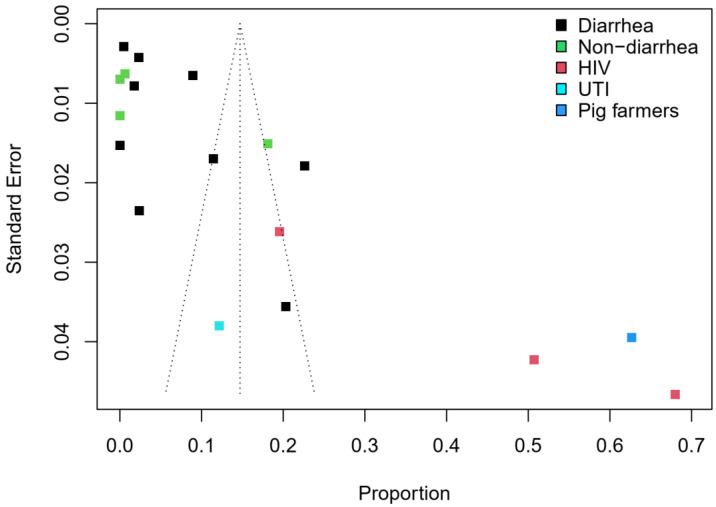
Funnel plot with 95% confidence limits showing the prevalence of *Campylobacter* species in humans in West Africa.

**Figure 7 pathogens-11-00140-f007:**
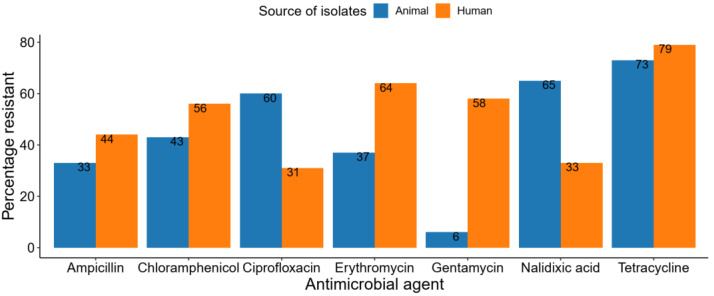
The proportion of *Campylobacter* spp. resistant to commonly tested antibiotics.

**Table 1 pathogens-11-00140-t001:** Pooled prevalence of *Campylobacter* spp. in animals stratified by subgroup variables.

Variables	Included Studies	Sample Size	Pooled Prevalence (95% CI)	*I*^2^ (%)	*p* Value
Country					
Nigeria	13	5702	34 (21–51)	99	<0.01
Ghana	6	1917	21(14–30)	94	<0.01
Burkina Faso	3	355	27 (5–73)	98	<0.01
Cote d’Ivoire	3	791	74 (52–88)	97	<0.01
Benin	1	256	33 (27–39)	-	-
Study setting ^a^					
Market	13	2367	37 (23–52)	97	<0.01
Farm	10	3955	31 (18–47)	99	<0.01
Abattoir	6	2670	33 (15–57)	99	<0.01
Veterinary clinic	1	473	11 (6–17)	-	-
Type of Sample ^a^					
Carcasses	13	3353	35 (21–53)	98	<0.01
Rectal swab	7	2930	33 (17–54)	98	<0.01
Feces	7	1719	32 (19–50)	97	<0.01
Preputial scraping	3	1122	20 (12–31)	92	<0.01
Diagnostic method					
Culture and biochemistry	16	5970	32 (21–47)	99	<0.01
Culture and PCR	5	1399	54 (28–78)	98	<0.01
PCR only	3	1106	22 (12–36)	88	<0.01
Culture and latex agglutination	1	346	43 (38–48)	-	-
Culture and MALDI-TOF MS	1	200	11 (7–16)	-	-
*Campylobacter* species					
*C. jejuni*	22	3075	52 (42–63)	96	<0.01
*C. coli*	17	2512	30 (22–40)	95	<0.01
*C. lari*	7	1420	12 (6–22)	84	<0.01
*C. fetus*	5	434	8 (1–46)	93	<0.01
*C. hyointestinalis*	4	505	4 (2–7)	39	0.18
*C. jejuni subsp.doylei*	3	320	5 (1–21)	80	<0.01
*C. upsaliensis*	2	292	12 (2–49)	89	<0.01
*C. sputorum*	1	36	6 (1–20)	-	-

*I*^2^-heterogeneity; ^a^ number of included studies is greater than 26 because three studies had data on two groups.

**Table 2 pathogens-11-00140-t002:** Pooled prevalence of *Campylobacter* spp. in humans stratified by subgroup variables.

Variables	Included Studies	Sample Size	Pooled Prevalence (95% CI)	*I*^2^ (%)	*p* Value
Country					
Nigeria	4	1182	22 (5–58)	98	<0.01
Ghana	3	1576	27(13–36)	97	<0.01
Burkina Faso	3	1729	2 (2–3)	0	0.45
Benin	1	45	1 (0–2)	-	-
Gambia	1	1933	9 (8–10)	-	-
Niger	1	350	11 (8–15)	-	-
Study setting					
Hospital	11	6620	10 (5–18)	98	<0.01
Community	2	195	14 (0–96)	92	<0.01
Study design					
Cross sectional	8	2618	19 (7–42)		99
Case control	4	2264	2(0–19)		96
Retrospective	1	1933	9 (8–10)	-	-
Age range					
Adults only (>15)	4	515	33 (13–62)	96	<0.01
<13 years	1	1234	20 (17–22)	-	-
<5 years	4	3268	4 (2–8)	93	<0.01
All ages	4	1798	9 (1–47)	99	<0.01
Diagnostic method					
Culture and biochemistry	6	2278	22 (7–51)	99	<0.01
PCR only	4	3412	7 (3–15)	97	<0.01
Culture and PCR	3	1125	4 (0–63)	99	<0.01
*Campylobacter* species					
*C. coli*	7	397	47 (25–69)	91	<0.01
*C. jejuni*	6	565	42 (26–59)	86	<0.01
*C. lari*	3	249	12 (4–28)	81	<0.01
*C. upsaliensis*	3	243	11 (3–33)	87	<0.01
*C. fetus*	2	165	13 (9–20)	0	0.61
*C. hyointestinalis*	2	139	6 (3–11)	0	0.75
*C. jejuni subsp.doylei*	1	35	3 (0–18)	-	-

*I*^2^-heterogeneity.

## Data Availability

The data presented in this study are available on request from the corresponding author.

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
