# Peer review of "Prevalence and Antibiotic Resistance in Campylobacter spp. Isolated from Humans and Food-Producing Animals in West Africa: A Systematic Review and Meta-Analysis"

_pathogens, 2022, doi:10.3390/pathogens11020140_

Round 1

Reviewer 1 Report

The objective of this work is very valuable. In countries where there are no surveillance systems for campylobacter, this type of work offers the scientific community a clear and transparent picture, the results of which can be subjected to interpretation by experts. The systematic review together with the data analysis has a solid scientific structure, starting from the first selection of scientific literature up to the criteria used for the final choice of sources. The analytical approach is also valuable through the use of R for pooled analysis. This reviewer also greatly appreciated the attention to the different variables considered by the authors, for example: the calculation of the error, the heterogenicity and the type of diagnostic investigation used in the various papers examined (PRC, Culture etc). Good relevance and completeness. 

Reviewer 2 Report

Line 77: Should read “Twenty eight of the full-text articles were excluded with reasons:” instead of “Eighteen of the full-text articles were excluded with reasons”

Figure 2: Please cite the source of the map used in the figure

Figure 5: Diarrheoa in figure and diarrhea in legend line 174. Please be consistent in the spelling

Line 185-186 Campylobacter spp. Prevalence in this study is higher in adult than kid less than 5 years old. Please provide a hypothesis It is usually the reverse.

Figure 7: please indicate in the figure if differences between sources are statistically significant.

Line 263-264: cdtA, cdtB, cdtC, cadF [6, 26]; Ser22Gly, Thr86Ile, 263 Asn203Ser [16] and tet (O), blaOXA-61, aadE, and cmeB genes [49]. That sentence doesn't make sense at all. cdtABC is not involved in antibiotic resistance. Please review the literature and explain clearly what mechanisms of Campylobacter are responsible for resistance including gyrA, rRNA, efflux pump, etc...

Line 293. Remove “t” before period.

Line 294-295: This could be also due to technical (i.e PCR Vs micro testing). Protocol used, staff training to isolate/found campylobacter, etc…

Figure 7 and discussion: the level of ciprofloxacin higher in animals Vs humans is puzzling. Cipro is not a commonly veterinary-used antibiotic. Please provide an hypothesis. Erythromycin is commonly used in the veterinary field but not in humans. Could the high level of erythromycin resistance observed in humans be explained by the over usage of Azithromycin?

Reviewer 3 Report

This paper describes an important issue of campylobacter prevalence in humans and animals in West Africa by literature review and meta analysis. The study is conducted according to standard procedures, is well written and coherent. But unfortunately, it is quite repetitive on what is already known, and authors do cite these references e.g. on their own numbering references [8,13,14]. These are also very recent, from 2020-2021, and yet even though mostly covering all Africa or sub-Saharan Africa, there is not much additional details. 

One way of improving the manuscript would be to add literature information on the sources and magnitude of antimicrobials used in animals and humans in WA. Is there any statistics on the quantities? How common is antimicrobial use in animal feed? If there is no statistics, these could be your recommendations, ie. to create systems to get at least some idea how much antimicrobials are used in humans/animals. Based on your study, should certain animal growth promoters be banned? Should some human antimicrobials only used in hospitals? Comprehensive paragraph to the introduction and discussion, and also compare the results obtained from humans and animals with respect to MDR to antimicrobial consumption.

I would also like to see the people living with HIV antimicrobial resistance separately presented, also species distribution (jejuni/coli/others). Do we know anything about the genetic profile (wgs) of the strains in humans/animals and also people living with HIV?

I would also like to see your public health conclusions. Would you propose limiting use of antimicrobials in animal husbandary? Would you recommend that human antimicrobial use should be directed by antimicrobial testing? Would you recommend limiting use of antimicrobials in children? Is there antimicrobial use policies in these countries and does them MDR results reflect the policies (countries with an antimicrobial policy have less MDR compared to those with no policy?).

Please feel free to add any other aspects or public health recommendations that you might come to think of. 

minor comments:

- around l. 150, the percentages are wrong way around for humans and animals. 

Round 2

Reviewer 3 Report

Dear Authors,

One minor issue to add. My original comments were off in terms of line numbers, I had written that animal and human results were opposite in line 150, while this should refer to lines 250-255 and Figure 7. In Figure 7 and test in lines 250-255 the animal and human referencing is the opposite. Please ensure they are rectified, and that those presented in the figure are correct. Otherwise, good to go. Nice job.

Author Response

This manuscript is a resubmission of an earlier submission. The following is a list of the peer review reports and author responses from that submission.